# Pandemic trends in health care use: From the hospital bed to self-care with COVID-19

**Fredrik Methi**[1]ʘ*, **Kjersti Helene Hernæs**[1]ʘ, **Katrine Damgaard Skyrud**[1]ʘ, **Karin Magnusson**[1,2]ʘ

**1** Cluster for Health Services Research, Norwegian Institute of Public Health, Oslo, Norway, **2** Clinical Epidemiology Unit, Orthopaedics, Department of Clinical Sciences Lund, Faculty of Medicine, Lund University, Lund, Sweden

ʘ These authors contributed equally to this work.
* fredrik.methi@fhi.no

**Data Availability Statement:** The study method and statistical analyses are all described in detail in the Methods chapter and throughout the paper. Individual-level data of patients included in this paper after de-identification are considered

## Abstract

### Aim

To explore whether the acute 30-day burden of COVID-19 on health care use has changed from February 2020 to February 2022.

### Methods

In all Norwegians (N = 493 520) who tested positive for SARS-CoV-2 in four pandemic waves (February 26th, 2020 –February 16th, 2021 (1st wave dominated by the Wuhan strain), February 17th–July 10th, 2021 (2nd wave dominated by the Alpha variant), July 11th–December 27th, 2021 (3rd wave dominated by the Delta variant), and December 28th, 2021 – January 14th, 2022 (4th wave dominated by the Omicron variant)), we studied the age- and sex-specific share of patients (by age groups 1–19, 20–67, and 68 or more) who had: 1) Relied on self-care, 2) used outpatient care (visiting general practitioners or emergency ward for COVID-19), and 3) used inpatient care (hospitalized ≥24 hours with COVID-19).

### Results

We find a remarkable decline in the use of health care services among COVID-19 patients for all age/sex groups throughout the pandemic. From 83% [95%CI = 83%-84%] visiting outpatient care in the first wave, to 80% [81%-81%], 69% [69%-69%], and 59% [59%-59%] in the second, third, and fourth wave. Similarly, from 4.9% [95%CI = 4.7%-5.0%] visiting inpatient care in the first wave, to 3.6% [3.4%-3.7%], 1.4% [1.3%-1.4%], and 0.5% [0.4%-0.5%]. Of persons testing positive for SARS-CoV-2, 41% [41%-41%] relied on self-care in the 30 days after testing positive in the fourth wave, compared to 16% [15%-16%] in the first wave.

### Conclusion

From 2020 to 2022, the use of COVID-19 related outpatient care services decreased with 29%, whereas the use of COVID-19 related inpatient care services decreased with 80%.

sensitive and will not be shared. However, the individual-level data in the registries compiled in Beredt C19 are accessible to authorized researchers after ethical approval and application to "helsedata.no/en" administered by the Norwegian Directorate of eHealth. Data requests may be sent to "service@helsedata.no.

**Funding:** The author(s) received no specific funding for this work.

**Competing interests:** The authors have declared that no competing interests exist.

## Introduction

The clinical course and mortality of COVID-19 among hospitalized individuals have been well described, especially for the start of the pandemic (spring 2020) [1–4]. Less is known about COVID-19's *total* use of health care services, including both inpatient and outpatient care both during the 1st wave, but especially during the 2nd, 3rd, and 4th waves striking spring and fall 2021 in Norway. To date, studies of outpatient care use during the pandemic have focused on structural changes in its delivery, such as telemedicine visits vs. office-based visits [5], whereas studies of the use of outpatient care services (such as general practitioners and emergency wards) are scarce. The use of health care services may be hypothesized to have changed throughout the pandemic, starting with limited test availability and many persons in risk groups being hospitalized and eventually dying, to mass testing, mass vaccination of persons at risk and an increasing herd immunity. In the latest wave, the health care services may be hypothesized to be better trained in how to manage severely ill patients, leading to declining inpatient care treatment and fewer fatal outcomes in the end than in the beginning of the pandemic.

A timely and correct up- and downscaling of health services (and lockdown measures) depend on our understanding of the pathways patients take through the health system, including the peaks and total demand of health care services following an individual's positive test. Studying the impact on both inpatient and outpatient care in its early waves, can provide valuable insight into health service needs in later stages, and contribute to the knowledge base that can increase our resilience against future pandemics.

We have access to high quality and very recent register data covering the first three pandemic waves, as well as the first weeks of the fourth wave. Thus, in this paper we aimed to explore the age- and sex-specific acute burden of COVID-19 on the health care services in four waves of the pandemic in Norway through a national descriptive cohort study design using registry data. Because of previously reported differences in vaccination status throughout the pandemic [6], differences in disease severity by SARS-CoV-2 variant [7, 8] and strata of age and sex [9], we hypothesized that we would also see differences in the 30-day pattern of healthcare use for men and women, girls and boys, the working age population, and the elderly in the different waves of the pandemic.

## Materials and methods

The BeredtC19-register is an emergency preparedness register aiming to provide rapid knowledge about the pandemic, including impacts of measures to limit the spread of the virus on health and utilization of health care services [10]. BeredtC19 compiles daily updated individual-level data from several registers. It includes the Norwegian Surveillance System for Communicable Diseases (MSIS) (all testing for COVID-19), the Norwegian Patient Register (NPR) (all electronic patient records from all hospitals in Norway), and the Norway Control and Payment of Health Reimbursement (KUHR) Database (all consultations with all general practitioners and emergency outpatient health care), as well as the National Population Register (age, sex, country of birth, date of death). Thus, the register includes all polymerase chain reaction (PCR) tests for SARS-CoV-2 in Norway with date of testing and test result, reported from all laboratories in Norway and all electronic patient records from primary care as well as outpatient and inpatient specialist care. The establishment of an emergency preparedness register forms part of the legally mandated responsibilities of The Norwegian Institute of Public Health (NIPH) during epidemics. Overall, data from Norwegian health registers have been demonstrated to be of high quality with high validity and reliability, and together they can provide a complete picture of patterns of healthcare use [11–13]. Medical recording to the National

registries is mandated by law in Norway, ensuring no missing data in our study. The Ethics Committee of South-East Norway confirmed (June 4th, 2020, #153204) that external ethical board review was not required.

## Population

Our population included every Norwegian resident who tested positive for the SARS-CoV-2 by a PCR-test from February 26th, 2020, to January 14th, 2022. The date with the first record of a confirmed test was coded as being the start of the individual's health care pathway. Patients with negative PCR-tests, as well as patients with suspected COVID-19 and without positive PCR-tests were excluded. For persons testing positive multiple times we included a wash-out period of 90 days [14]. We divided our population into mutually exclusive age and sex groups, i.e., girls and boys, men, and women by the following age categories: 1–19 (children and adolescents), 20–67 (working age population) and 68 years or older (elderly), as COVID-19 has hit differently among different age groups and sexes [9].

## Outcomes

Our outcomes were defined as follows:

1.  Self-care: No registered health care use (outpatient or inpatient) within 30 days of a positive test.

2.  Outpatient care: Outpatient care use with International Classification of Primary Care (ICPC-2) code R991 or R992 (COVID-19) (general practitioners or emergency wards).

3.  Inpatient care: Hospital-based inpatient specialist care with International Classification of Disease (ICD-10) code U071 (confirmed COVID-19) or U072 (suspected COVID-19).

4.  Death: Death independent of cause but occurring within 30 days after the positive test.

The dates of all outcomes were sorted relative to the date of the positive PCR-test, with the test date being coded as day 0 and the outcomes occurring on day -2 to day 30. We chose a 30-days-timeframe because a death after COVID-19 was classified as covid-related if it occurred within 30 days after testing positive in official statistics [15], and because it coincides with what is commonly referred to as the acute phase of SARS-CoV-2 [16]. Thus, we regarded people who were still alive after 30 days as recovered. When combined, and sorted chronologically on dates of occurrence, these data provided a comprehensive picture of the peak and total use of outpatient and inpatient care, as well as COVID-19-related health care pathways in the acute phase. To account for the fact that many PCR-tests were prescribed by the health care services, we ran analyzes both including and excluding health care use related to the testing.

## Study setting

The COVID-19 epidemic in Norway has, since the first registered case on February 26th, 2020, been dominated by four different lineages (Fig 1). In this article we define four waves of the pandemic based on the dominating virus lineages. The 1st wave of transmission (February 26th, 2020 –February 16th, 2021) was characterized by low test availability (only available for health personnel, elderly, and persons at risk) until the summer of 2020, and was primarily dominated by the Wuhan strain of the virus as well as some minor cases of other lineages such as Beta and Gamma. The 2nd wave of transmission (February 17th, 2021 –July 10th, 2021) was characterized by wide testing criteria and free testing, start of vaccination, and was dominated by the Alpha variant. The 3rd wave of transmission (July 11th, 2021 –December 27th, 2021) was

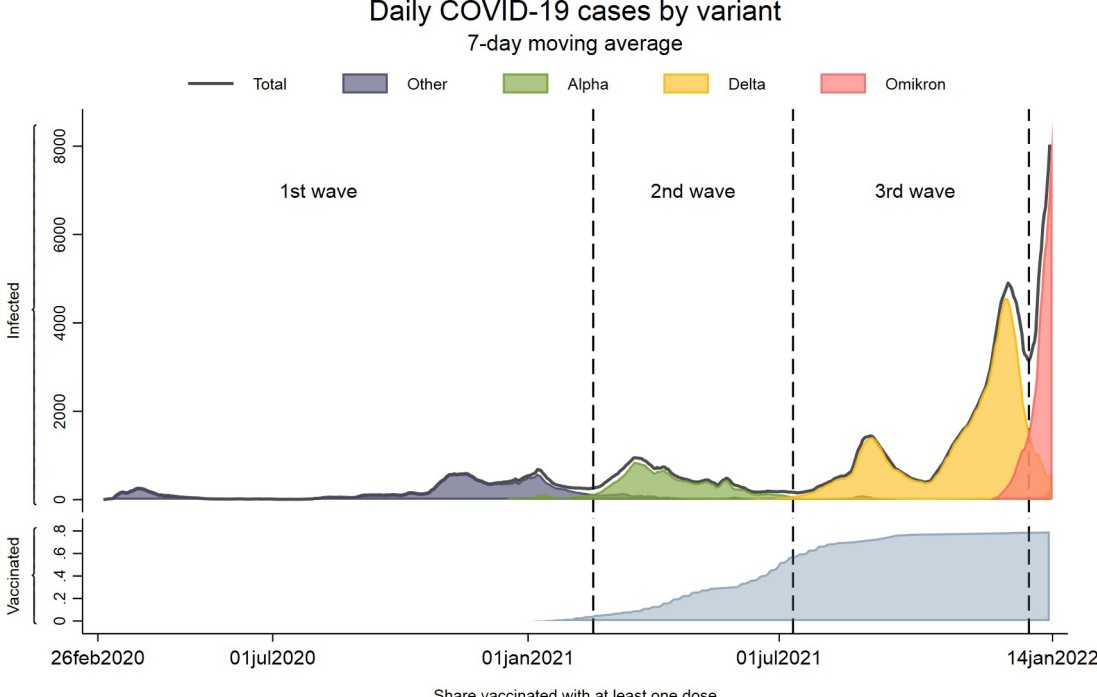

**Fig 1. Daily covid-19 cases by variant and share of population vaccinated.** The top panel shows the number of persons with a positive PCR-test each day calculated with a 7-day moving average. The solid line in the top panel shows the total numbers of persons infected, and colors represent the share of screened or sequenced tests with each variant the given day. The bottom panel shows the cumulative share of the included population having received at least one dose of SARS-CoV-2 vaccine on the given day.

characterized by continued wide testing criteria and free testing, most people vaccinated with at least one dose and was dominated by the Delta variant. We also included data from the 4th wave of transmission (December 28th, 2021 –January 14th, 2022), which was dominated by the Omikron variant and saw start of booster vaccination [17–19] (Fig 1).

## Statistical analyses

In this national descriptive cohort study using registry data, we first assessed descriptive statistics of our study population by the four different waves or periods of the pandemic.

Second, we studied the peak and total use of outpatient and inpatient care during the -2 to +30 days following positive test, for each of the waves. To explore the wave-wise *peak use*, we estimated day-by-day proportions in need of outpatient or inpatient care at least once during day -2 to +30. For persons with at least one outpatient care visit we included a maximum of one visit per day, and for persons with at least one inpatient care visit, we coded all the days spent in hospital as hospital bed-days. To explore the *total use* in each wave we estimated the cumulative proportions seeking outpatient or inpatient care at least once during day -2 to +30. We repeated these analyses, excluding day –2 to 0 to account for the impact of testing at the peak and total use of health care. Among persons seeking outpatient or inpatient care at least once, we estimated the mean number of outpatient visits or the mean number of hospital bed days in each wave, by age and gender.

Finally, based on the total health care use observed during the -2 to +30 days following positive test, we divided the study population into three different mutually exclusive major patient pathways. Each of the patient pathways represented a different acute burden of disease on the

health care systems: 1) persons relying on self-care with no health care use, 2) patients who had contact with outpatient care (GP and/or emergency ward) only, and 3) patients who had contact with inpatient care, with or without additional need for outpatient care. For each age-/sex-group and for each of their pathways in each of their pandemic wave, we estimated the proportions having the different pathways and calculated 95% confidence intervals. To get an overarching picture of the major patient flows, we visualized the timing of care for these different pathways in alluvial diagrams. We also estimated the whole-sample- and pathway-specific mortality as proportions with 95% confidence intervals. 95% confidence intervals were calculated as $\mu \pm 1.96 \times \frac{\sigma}{\sqrt{n}}$ where $\mu$ is the mean, $\sigma$ is the standard deviation and n is the population size. All analyses were run using STATA SE v.16.

## Results

We identified 493 520 persons with at least one positive PCR-test for SARS-CoV-2 in the total tested population of 3 625 617 persons between February 26th, 2020, and January 14th, 2022. The total number of tests throughout the pandemic was 10 520 144. The percentages of positive tests among all tests in the 1st, 2nd, 3rd, and 4th waves were 1.9%, 2.2%, 8.0%, and 27.2%, respectively. Table 1 shows that the average age of persons testing positive decreased from the 1st to the 4th wave. It also shows that the percentage of women among those testing positive increased from the first two waves to the last two waves (Table 1). The proportions dying within 30 days after positive test decreased throughout the pandemic, from 11.3% to 1.8% for persons above 68 years (S1 Table). The 30-day mortality was low across all pandemic waves for persons aged under 67 years (S1 Table).

**Table 1. Descriptive characteristics of persons testing positive for SARS-CoV-2 in each of four pandemic waves in Norway, 2020–2022.**

| | | 1st wave | 2nd wave | 3rd wave | 4th wave |
|---|---|---|---|---|---|
| | | Feb 26th '20—Feb 16th '21 | Feb 17th '21 –Jul 10th '21 | Jul 11th '21—Dec 27th '21 | Dec 28th '21 –Jan 14th '22 |
| | | N = 64 254 | N = 64 181 | N = 244 517 | N = 120 568 |
| Characteristics of the whole sample | | | | | |
| Age, mean (SD) | | 37.0 (19.5) | 30.2 (17.9) | 28.7 (19.5) | 28.4 (17.5) |
| Women, N (%) | | 30533 (47.5) | 30107 (46.9) | 120531 (49.3) | 59779 (49.6) |
| Outcomes | | | | | |
| | Self-care (%) | 9941 (15.5) | 12039 (18.8) | 75,242 (30.8) | 49,216 (40.8) |
| | Outpatient care (%) | 53549 (83.3) | 51850 (80.8) | 168200 (68.8) | 71107 (59.0) |
| | Inpatient care (%) | 3129 (4.9) | 2294 (3.6) | 3329 (1.4) | 570 (0.5) |
| | All-cause mortality (%) | 590 (0.9) | 146 (0.2) | 571 (0.2) | 52 (0.0) |
| Share vaccinated by start of the wave (min. 1 dose) | | 0% | 5% | 57% | 79% |
| Characteristics of the studied strata | | | | | |
| Children and adolescents (1–19 years) | | | | | |
| | Girls, N (%) | 6119 (9.5) | 10601 (16.5) | 49988 (20.4) | 21558 (17.9) |
| | Boys, N (%) | 6521 (10.1) | 11051 (17.2) | 51980 (21.3) | 22049 (18.3) |
| Adults in working age (20–67 years) | | | | | |
| | Women, N (%) | 21949 (34.2) | 18702 (29.1) | 65883 (27.0) | 36996 (30.7) |
| | Men, N (%) | 25031 (39.0) | 22133 (34.5) | 67387 (27.6) | 37553 (31.2) |
| Elderly (≥68 years) | | | | | |
| | Women, N (%) | 2449 (3.8) | 792 (1.2) | 4654 (1.9) | 1225 (1.0) |
| | Men, N (%) | 2185 (3.4) | 902 (1.4) | 4625 (1.9) | 1187 (1.0) |

## Peak and total use of outpatient care

In the early phases of the pandemic, patients sought outpatient care frequently. 83% of all who tested positive sought outpatient care within 30 days in the 1st wave. This share decreased only marginally for the 2nd wave (81%), and decreased further to 69% in the 3rd wave, and to 59% in the 4th wave. During the 1st wave, 40% of all patients sought outpatient care on day 0 (including day -1 and -2). This day-0-proportion decreased to ~30% in the 2nd and 3rd wave, and further to 25% in the 4th wave (Fig 2). Furthermore, during the 1st wave, patients continued to visit outpatient care for a longer time compared to the other waves, with 2.4% still visiting outpatient care on day 20, compared to 1.5% in the 2nd wave, 0.6% in the 3rd wave, and 0.3% in the 4th wave (Fig 2). S2 Fig shows that middle-aged and elderly persons continued to seek outpatient care for a longer time than persons aged 1–19 years. For all age groups, the mean number of visits (among those seeking outpatient care) steadily decreased for each wave of the pandemic, from ranging between 2–4 visits per patient, to 1–2 per patient in the 4th wave (Fig 3). When we excluded outpatient care use in relation to testing, results showed no change in the cumulative share from the 1st wave (76%) to the 2nd wave (75%), a slight decrease in the 3rd wave (58%) and continued decrease in the 4th wave (45%) (S3 Fig).

## Peak and total use of inpatient care

We observed an even more considerable shift in the total use of inpatient care throughout the pandemic. From 4.9% of all who tested positive being hospitalized at least once during the 1st wave, compared to 3.6% during the 2nd wave, 1.4% in the 3rd wave, and 0.5% in the 4th wave (Fig 4). For the three first waves, the share seeking inpatient health care peaked between the 8th and 11th day following a positive test (Fig 4). For the fourth wave, the peak was observed on

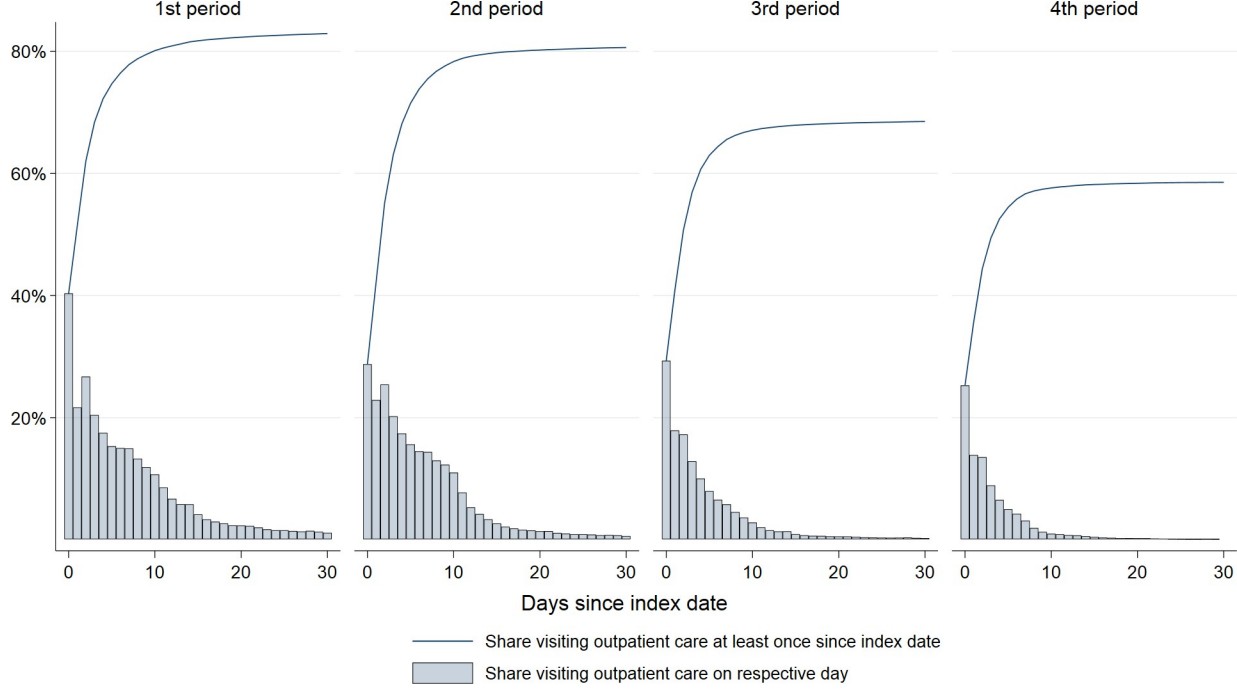

**Fig 2. Shares visiting outpatient care.** Day by day and cumulative shares seeking outpatient care (GP or emergency ward) from the day of testing positive (day 0) to the 30th day after positive test, for four waves/periods of the pandemic: February 26th, 2020 –February 16th, 2021 (1st period), February 17th, 2021 –July 10th, 2021 (2nd period), July 11th, 2021 –December 27th, 2021 (3rd period), and December 28th, 2021 –January 14th, 2022 (4th period).

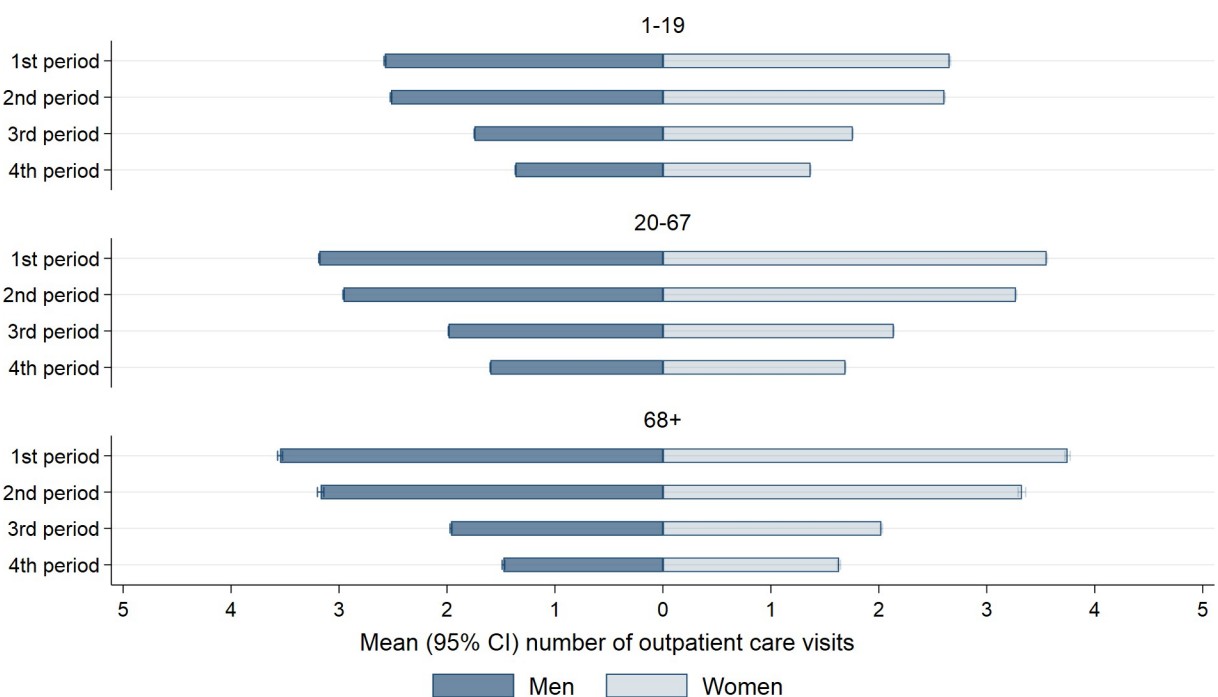

**Fig 3. Number of outpatient care visits.** The mean (95% confidence intervals) number of visits in outpatient care, for women and men in different age groups, four waves/periods of the pandemic: February 26th, 2020 –February 16th, 2021 (1st period), February 17th, 2021 –July 10th, 2021 (2nd period), July 11th, 2021 –December 27th, 2021 (3rd period), and December 28th, 2021 –January 14th, 2022 (4th period), among persons having at least one visit in outpatient care.

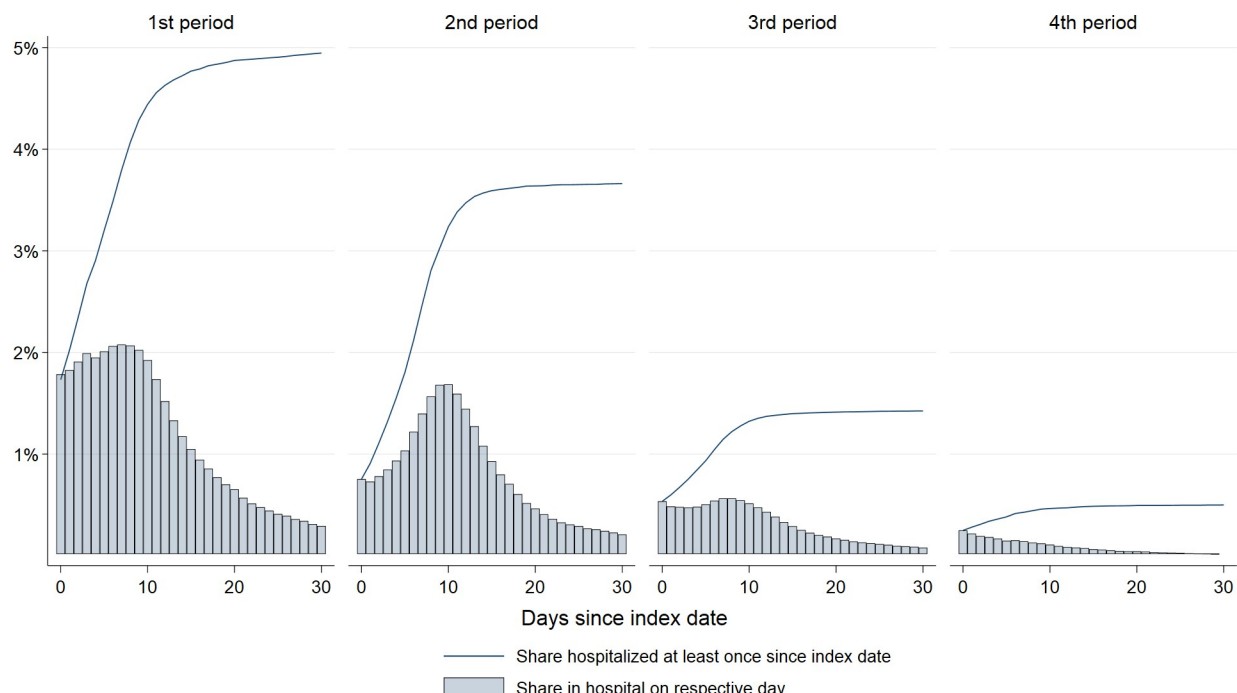

**Fig 4. Shares seeking inpatient care.** Day by day and cumulative shares seeking inpatient care from the day of testing positive (day 0) to the 30th day after positive test, during four waves/periods of the pandemic: February 26th, 2020 –February 16th, 2021 (1st period), February 17th, 2021 – July 10th, 2021 (2nd period), July 11th, 2021 –December 27th, 2021 (3rd period), and December 28th, 2021 –January 14th, 2022 (4th period).

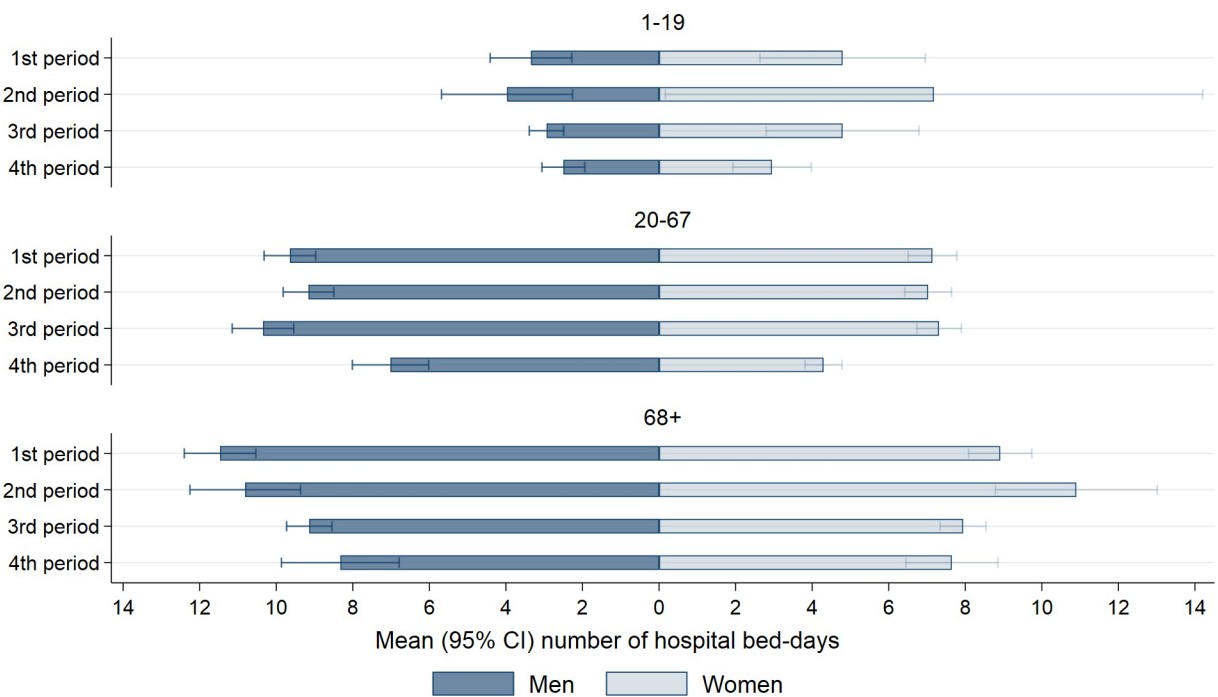

**Fig 5. Number of bed-days spent in inpatient care.** The mean (95% confidence intervals) number of bed-days spent in inpatient care, for women and men in different age groups, during four waves/periods of the pandemic: February 26th, 2020 –February 16th, 2021 (1st period), February 17th, 2021 –July 10th, 2021 (2nd period), July 11th, 2021 –December 27th, 2021 (3rd period), and December 28th, 2021 –January 14th, 2022 (4th period), among persons having at least one visit in inpatient care.

the first day. S4 Fig shows that a minor proportion of children and adolescents, and a considerable proportion of elderly sought inpatient care. Along this line, the mean number of bed-days was also higher for the middle-aged and elderly, than it was for children (Fig 5). Generally, women tended to have a lower number of bed-days than men (Fig 5). For men and women in their working age (20–67 years), and for men aged ≥68 years the mean number of bed-days decreased from the first to the fourth wave (Fig 5). No statistically significant decrease was observed for children or for women aged 68 or more (Fig 5). As expected, we see little changes in the cumulative share of persons seeking inpatient care when excluding days –2 to 0, i.e., during the period in which any health care use could be assumed to be related to testing (S5 Fig).

## Patient pathways from positive test to 30 days after

Overall, both men and women in all age groups largely followed the same patient pathways in the first two waves (Fig 6). The share seeking outpatient care decreased for those aged 1–19 (from 77% [95%CI = 76%-78%] to 63% [63%-64%]) and 20–67 (from 79% [78%-82%] to 72% [71%-72%]) from the first two waves to the third wave, respectively, while this remained stable from the first two waves to the third wave for those aged 68+ (57% [55%-59%]). In the fourth wave, the decrease in outpatient health care use continued for those aged 1–19 (51% [51–52]), 20–67 (63% [62%-63%]), and now also for those aged 68+ (50% [48%-52%]) (Fig 6).

Along this decrease in outpatient care use, we also observed a decrease in the share of persons in need for inpatient care, especially from the 2nd to 4th wave for all groups of age and sex (Fig 6). For patients aged 68+ the share hospitalized declined from 25% [23%-27%] the first two ways, to 7% [5%-8%] in the fourth wave. The share that did not use any health care services (neither inpatient or outpatient) was relatively stable in wave 1 and 2 for all age groups and both sexes, but increased in wave 3, and continued to increase in wave 4 (Fig 6).

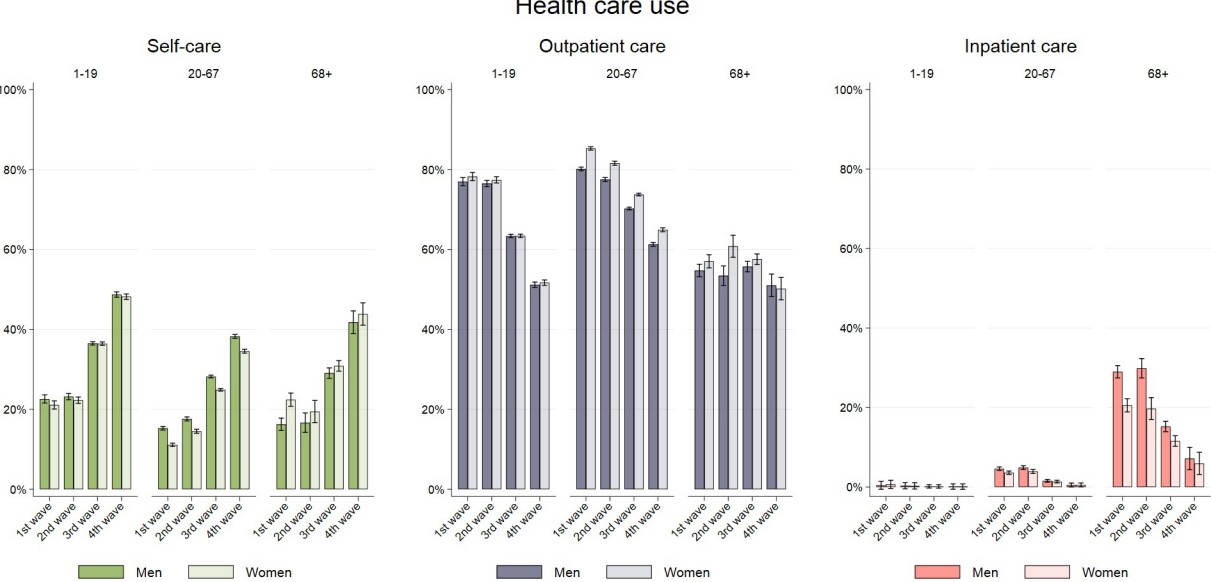

**Fig 6. Health care use.** Health care use during the first 30 days after a positive PCR-test for SARS-CoV-2 by age groups and gender, for four waves/periods of the pandemic: February 26th, 2020 –February 16th, 2021 (1st period), February 17th, 2021 –July 10th, 2021 (2nd period), July 11th, 2021 –December 27th, 2021 (3rd period), and December 28th, 2021 –January 14th, 2022 (4th period).

The shares that were still in need of outpatient or inpatient care on day 30 after a positive test decreased from the 1st to the 2nd wave and to a lesser extent from the 2nd to the 3rd wave, for all groups of age and sex. During the 1st wave, a smaller shared of those aged ≥68 years relied on self-care on days 21–30 after a positive test, compared to what is seen during the 3rd wave and 4th wave (S1 Fig).

And finally, the overall 30-day all-cause mortality decreased for all persons from the 1st to the 4th wave (S1 Table).

## Discussion

In this descriptive study of all 493 520 persons testing positive for SARS-CoV-2 from February 26th, 2020 to January 14th, 2022 in Norway, we find that the share of patients with COVID-19 in need for health care services in the acute phase (30 days) has decreased throughout the pandemic, from 83% [95%CI = 83%-84%] to 59% [59%-59%] for outpatient care and from 4.9% [95%CI = 4.7%-5.0%] to 0.5% [0.4%-0.5%] for inpatient care. Accordingly, the share relying on self-care only increased, from 16% [15%-16%] in the first wave to 41% [41%-41%] in the fourth wave.

### Comparison with previous studies

To our knowledge, this study is the first to describe COVID-19 related patterns in both inpatient and outpatient care use following positive test for SARS-CoV-2. The massive use of outpatient care services (including visits to the general practitioner and emergency wards) following COVID-19 has not previously been reported, i.e., no knowledge foundation exists for an effective comparison of findings across studies.

Far more is known for inpatient care use. Compared to previous studies we find, on average, lower hospitalization rates in Norway than what has been reported in other countries [20–22]. As an example, Denmark and Sweden observed hospitalization rates of 20% and 16%,

respectively, in the first wave of the pandemic [21, 22], which contrasts with Norwegian hospitalization rates of only 5% (Table 1). The differences may be explained by differences in test criteria, but also differences in criteria for admission to the hospital. Further, the Danish and Swedish rates are based on a pandemic period that was shorter than studied in the current study [21, 22]. There were also differences in mortality between the Scandinavian countries. Whereas all-cause mortality among COVID-19 patients in Sweden was observed to be around 4–5% [22], we find a mortality rate of only 0.9% in Norway between February 2020 and February 2021 (Table 1). Again, the discrepancies might be explained by the difference in length of the study periods as well as differences in registration practices of deaths. However, in line with previous studies, we see higher mortality rates among men compared to women, particularly for those younger than 68 years [10]. Also, in line with previous reports, we find that a significant share of those who died, died within the first 10 days (S1 Fig) [23].

Another important observation to our inpatient care results, was the tendencies that the mean number of bed-days in hospital increased for some age- and sex-groups from the 1st to the 2nd wave (Fig 5). Although we did not aim to explore whether the severity of COVID-19 has changed, an important characteristic of the 2nd, 3rd, and 4th waves of transmission has been the rise of mutant viruses. Reports have been inconclusive as to whether mutant viruses (Alpha/Beta/Gamma vs. Wuhan strain) result in more severe disease requiring more hospital care [19, 24, 25]. Whilst the risk of hospitalization between the Alpha and Delta variant were found similar, the Omicron variant has had a reduced risk of hospitalization compared to the Delta variant [8, 9].

### Interpretation and relevance

The high burden put on outpatient care services is important to report, given that a well-functioning outpatient care service is essential in reducing demands put on hospital services; it is essential to support rehabilitation of recovering patients; to improve palliative care; and sustain non-covid care [26]. As an example, S1 Fig shows that a larger share used inpatient care prior to outpatient care, than the other way around, suggesting that outpatient care to some extent has been used for recovery issues after hospitalization. As such, more knowledge of rehabilitation of COVID-19 patients (severely and mildly affected) in the community care services is needed.

Still, in our study, the peak use of outpatient care was centered to the -2 to 0 days around positive test, implying that a certain proportion of the large amount of outpatient care visits took place in relation to testing and the detection of COVID-19. Indeed, when we excluded visits that were related to testing, the total share visiting outpatient care during the 30-day period decreased, yet only slightly (from ~80% to ~70%). Thus, the somewhat different patterns in outpatient care use from the 1st and 2nd to the 3rd and 4th wave (Fig 2) may be explained by differences in testing criteria, i.e., the start of the 1st wave was the only wave with limited test availability and strict testing criteria (the elderly, persons at risk and health personnel). We believe that such differences in testing patterns are less likely to explain the decreasing demand put on inpatient care services throughout the pandemic (because frail and elderly persons were tested to an equal extent independent of pandemic wave).

A last interpretation of our findings is that healthcare workers in outpatient care may have a higher threshold for referral to inpatient care in 2022 than in 2020. This may be explained by increased knowledge of the disease and its outcomes (including lower mortality rates) as the pandemic progressed. The milder SARS-CoV-2 omicron variant and a higher vaccination coverage might also explain the observed shifts in health care use (Fig 1). However, we cannot exclude that more severe mutations in the future again place an increasing demand on the

health care services. Also, vaccines might have lower effects against new variants. Thus, if we again see SARS-CoV-2 variants that cause the same disease severity as the initial variants, our findings of an only halved outpatient care use but ten times lower inpatient care use from February 2020 to February 2022 suggest that an upscaling of the outpatient care services might be particularly important in the future.

## Strength and limitations

An important strength of our study is that we could include everyone with a positive test throughout four major waves or periods of the pandemic. In this way, we could provide a comprehensive picture of all health care use following a positive PCR-test for these different waves, i.e., not restricted to inpatient care as in previous studies [2, 3]. Moreover, we could provide details in outpatient and inpatient care for different age and sex groups.

Several limitations should be mentioned. First, we do not know the causes or severity of complaints behind the care use following a positive test for SARS-CoV-2. Although we only included care visits with diagnostic codes of COVID-19, we did not separate the complaints affecting e.g., the respiratory or digestive system. Also, we had no comparison group, simply because we did not aim for any causal inference and because comparable data are not available for a similar epidemic or pandemic setting with other infectious diseases. However, in recent studies of post-acute COVID-19, we demonstrate a likely causal effect of being infected with SARS-CoV-2 on the post-acute health care use [27]. Here, we also exclusively included visits that were specific to COVID-19, i.e., we did not study all-cause visits. Second, our study was of an explorative and descriptive character. Thus, we looked for patterns and trends in a large amount of data using mainly graphs in a self-developed structure, such as the division of age into children and adolescents, adults in working age population and the elderly, and by sex. We did not apply any data-driven analyses in our exploration of pandemic trends in health care use, thus we might have missed important details. To combat some of these issues, we chose to present a large amount of raw data visualized as alluvial diagrams in the S1 Fig. Third, we may have underestimated the care use among persons aged 68 years or more. Very frail persons live in care homes and receive institutionalized care that may not be registered in our data sources. And finally, and as mentioned above, we cannot exclude that some of our observations of changing (or stable) trends are due to differences in test criteria or -patterns as the pandemic progressed. Such patterns may differ across our groups of age and sex. However, if this is the case, the testing is obviously a part of health care use in relation to COVID-19, or else we would not have observed these visits. Thus, because testing for SARS-CoV-2 has been a part of the outpatient and inpatient care services from the beginning of the pandemic, including care visits in relation to the detection of COVID-19 is still important in the public health question of whether the health services should be upscaled or downscaled in future similar situations.

In conclusion, we demonstrate a decreasing impact of COVID-19 on all COVID-19 related health care services from the 1st to the 4th wave of the pandemic. The use of COVID-19 related outpatient care services was reduced with 29%, whereas the use of COVID-19 related inpatient care services was reduced with 80% in January-February 2022 compared to the first year of the pandemic. These findings are important to report considering future mutations and waves of COVID-19, i.e., there may be a lower need for upscaling inpatient care services and a large need for upscaling outpatient care services.

## Supporting information

**S1 Fig. Alluvial diagram showing patient-flows in ten day-intervals by age group and period.** Note: This figure visualizes the most common pathway-flows for each period and each

age-group. SC = Self-care; OC = Outpatient care; IC = Inpatient care; and Dead = Dead within 30 days. Colors show which strata each person belonged to in the last 10 days. Categories are mutually exclusive and ordered by severeness (Death > IC > OC > SC). Crude numbers are removed in order to avoid confusion. The sizes of each stratum should therefore be interpreted as shares.
(PDF)

**S2 Fig. Share visiting outpatient care withing 30 days of testing positive by age and sex.** Note: Day 0 includes 0–2 days before testing positive.
(PDF)

**S3 Fig. Day by day cumulative share visiting outpatient care (GP or emergency ward) from one day after testing positive (day 1) to 30th day after positive test, i.e., excluding outpatient care visits that were related to the testing and detection of SARS-CoV-2.**
(PDF)

**S4 Fig. Share visiting inpatient care withing 30 days of testing positive by age and sex.** Note: Day 0 includes 0–2 days before testing positive.
(PDF)

**S5 Fig. Day by day cumulative share visiting inpatient care from one day after testing positive (day 1) to 30th day after positive test, i.e., excluding inpatient care visits that were related to the testing and detection of SARS-CoV-2.**
(PDF)

**S1 Table. Outcomes of persons testing positive for SARS-CoV-2 within 30 days, in each of four pandemic waves in Norway, 2020–2022.** Note: Due to privacy reasons we cannot report exact numbers when numbers are between 0 and 5. Therefore we have censored the exact numbers for deaths in the 1st and 3rd wave for the various age groups. The table still includes the percentages and 95% confidence intervals.
(PDF)

## Acknowledgments

We would like to thank the Norwegian Directorate of Health, in particular Director for Health Registries Olav Isak Sjøflot and his department, for excellent cooperation in establishing the emergency preparedness register. We would also like to thank Gutorm Høgåsen and Anja Elsrud Schou Lindman for their invaluable efforts in the work on the register. We would also like to thank Kjetil Telle, Anja Elsrud Schou Lindman, Karin Maria Nygård and Gunnar Øyvind Isaksson Rø for critically evaluating the content of the study. The interpretation and reporting of the data are the sole responsibility of the authors, and no endorsement by the register is intended or should be inferred. We would also like to thank everyone at the Norwegian Institute of Public Health who has been part of the outbreak investigation and response team.

## Author Contributions

**Conceptualization:** Karin Magnusson.

**Data curation:** Fredrik Methi, Kjersti Helene Hernæs, Katrine Damgaard Skyrud.

**Formal analysis:** Fredrik Methi, Kjersti Helene Hernæs, Katrine Damgaard Skyrud.

**Methodology:** Fredrik Methi, Kjersti Helene Hernæs, Katrine Damgaard Skyrud.

**Supervision:** Karin Magnusson.

**Validation:** Fredrik Methi, Kjersti Helene Hernæs, Katrine Damgaard Skyrud, Karin Magnusson.

**Visualization:** Fredrik Methi, Kjersti Helene Hernæs, Katrine Damgaard Skyrud.

**Writing – original draft:** Karin Magnusson.

**Writing – review & editing:** Fredrik Methi, Kjersti Helene Hernæs, Katrine Damgaard Skyrud, Karin Magnusson.

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
