## [Decision Letter · Decision Letter 0]

10 Jan 2022

PONE-D-21-37160Pandemic trends in health care use: From the hospital bed to the general practitioner with COVID-19PLOS ONE

Dear Dr. Methi,

Thank you for submitting your manuscript to PLOS ONE. After careful consideration, we feel that it has merit but does not fully meet PLOS ONE’s publication criteria as it currently stands. Therefore, we invite you to submit a revised version of the manuscript that addresses the points raised during the review process. Please read carefully the reviewers' comments in order to address specifically the following points:

1-Describe the study design, make sure your paper follows the STROBE checklist for observational studies

2-Please address the comments of reviewer #2 regarding the BEREDT-C19 registry and its validation data. What are the flaws of the registry, missing data?

3-Clearly define the criteria for primary care and specialist care, as highlighted by reviewers #2 and #3.

4-Please explain the rationale behind using age- and sex-specific share of patients to assess healthcare system use and the rationale behind the 30-day acute phase as highlighted by reviewer #3

5-Clarify figure S1

6-Please use one term either share or fraction

We look forward to receiving your revised manuscript.

Kind regards,

Mabel Aoun, MD, MPH

Academic Editor

PLOS ONE

Journal Requirements:

Reviewers' comments:

Reviewer's Responses to Questions

**Comments to the Author**

1. Is the manuscript technically sound, and do the data support the conclusions?

Reviewer #1: Yes

Reviewer #2: Yes

Reviewer #3: Yes

2. Has the statistical analysis been performed appropriately and rigorously? 

Reviewer #1: Yes

Reviewer #2: Yes

Reviewer #3: Yes

3. Have the authors made all data underlying the findings in their manuscript fully available?

Reviewer #1: Yes

Reviewer #2: No

Reviewer #3: No

4. Is the manuscript presented in an intelligible fashion and written in standard English?

Reviewer #1: Yes

Reviewer #2: Yes

Reviewer #3: No

5. Review Comments to the Author

Reviewer #1: Thanks to the authors. I reviewed the article. It is valuable research in the pandemic era. The title, introduction, methodology, and conclusion are appropriate. There is no plagiarism in it. The writing is appropriate and completely understandable. I have minor recommendation as followings:

1- How did the authors delete the effect of confounding factors from the study outcome?

2- Discussion should be improved and revised based on the study results and comparison with previous ones i.e. Bagi HM, Soleimanpour M, Abdollahi F, Soleimanpour H (2021) Evaluation of clinical outcomes of patients with mild symptoms of coronavirus disease 2019 (COVID-19) discharged from the emergency department. PLoS ONE 16(10): e0258697. https://doi.org/10.1371/journal.pone.0258697

3- What is the new finding of this study compared to previous ones this field?

4- Please mention the weak and strong points of your study

5- The list of abbreviations at the end of the manuscript is absence; please make sure all your abbreviations are listed.

6- I recommend providing a table on the characteristics of your sample, including all variables you use in your analysis later on.

7- Please describe the study design in more detail.

8- provide more information on how you analyzed your data and possibly provide some references for e.g. level of statistical significance.

Reviewer #2: Introduction (pages 3-4): Please state study design (national cohort study?) and any prespecified hypotheses. More generally, please review and consider using STROBE checklist for observational studies (e.g. https://www.strobe-statement.org/checklists/).

Methods (page 4): An important feature of this study is its use of Norwegian national registries. The authors give one reference to a webpage that describes the BEREDT-C19 registry. I’m unable to find any validation data on that website. If such validation data exists, the authors should cite it directly. If validation data of the BEREDT-C19 registry does not exist but validation of the underlying registries from which BEREDT-C19 draws does exist, I would cite that data instead. The validity of these registries may be well known to Norwegian epidemiologists, but should be established for international readers.

Methods (page 5-6): It becomes apparent here that the authors are including treatment received in emergency wards as “primary care,” as contrasted with “specialist care” meaning hospital admission. I personally wouldn’t usually think of emergency department visits as part of primary care, and I don’t think most US physicians would either. Conversely, specialist care doesn’t clearly connote hospitalization to me (I would think of e.g. an outpatient cardiology clinic visit as “specialist care”). If I’m understanding the authors’ intentions correctly, I wonder if the manuscript might be clearer for international readers if it consistently referred to “outpatient care” (including clinic and emergency department visits) vs. “inpatient care” (hospital admission) rather than to “primary care” vs. “specialist care?”

Methods (page 6): the authors describe the three waves they are examining and mention the start of mass vaccination during the third wave. It would be helpful to have numbers and citations for the rates of vaccination at the beginning and end of this wave, since this would be expected to significantly impact rates of hospitalizations and I would think this data would be relatively easy to obtain.

Results (pages 7-11): The authors don’t have a comparison group but make many comparisons between waves, e.g. “we observed a significant shift in the total use of specialist care throughout the pandemic, from 14% being hospitalized at least once during the 1st wave, compared to 4% during the 2nd and 3rd waves (Fig 4).” (page 10) Consider measures of statistical significance or confidence intervals for differences.

Discussion (page 12): The authors state that “we were unable to find a similar study for an effective comparison of our findings.” I agree that I also am not able to find a similar study of the proportion of COVID patients receiving outpatient care. However, as the authors note, there are many studies of hospitalization rates among patients with COVID (e.g. Menachemi N, Dixon BE, Wools-Kaloustian KK, Yiannoutsos CT, Halverson PK. How Many SARS-CoV-2-Infected People Require Hospitalization? Using Random Sample Testing to Better Inform Preparedness Efforts. J Public Health Manag Pract. 2021 May-Jun 01;27(3):246-250. doi: 10.1097/PHH.0000000000001331. PMID: 33729203.). The authors should discuss how their findings regarding hospitalization rates compare to those found in other studies (the hospitalization rates the authors found appear to me higher than those found in some other studies).

Discussion (page 13): The authors note that the proportion of patients hospitalized and the duration of hospitalization increased from wave 2 to wave 3. Given that the authors state that vaccines became available during wave 3, these findings are surprising. Again, it would be helpful to know what proportion of the population was vaccinated at the beginning and end of the 3rd wave. Some comment might be offered in the discussion.

Finally, the authors may consider including data from the current omicron wave. I don’t think it’s absolutely necessary, and the publication need not and should not be delayed until the pandemic fully passes, but the study would be even more informative with data from this fourth wave included.

Reviewer #3: This study highlights a major topic of interest during the Covid 19 pandemic.

The healthcare burden imposed by the pandemic is one of the main factors to be studied in order to improve preparedness plans for the coming waves and to enhance knowledge about dealing with future pandemics. This study deals with exhaustive data from a national registry and is theoretically well positioned to give valuable information about the research question risen by the authors.

Understanding patients flow during the pandemic is another important facet of managing healthcare resources knowing that the study reports data from Norway, a country known for a very distinguished and organized health system.

The authors decided to study sex and age specific use of healthcare system, I would appreciate adding to the introduction the rationale behind this choice as they were only mentioned in the research question without any evidence to support this choice.

The 30 days’ time frame is also a choice made by the authors and showing evidence supporting this time limit -mean time to recovery, incidence of late onset complications…- would also be of value.

The authors developed in a descent way the importance of their research question. Developing the potential applications of such data would add value to the manuscript.

The outcomes measures are not well described in the methods section. Primary care use for example is a broad topic and the indicators used for measurement developed later may figure in a paragraph with a detailed description of the calculation methods used for every variable used to assess this outcome.

A review of the editing and English proofing would also be helpful to make the reading even more enjoyable.

Lines 59-72: the description of the registry is fair enough and gives a good insight of its contents and objectives. A clarification about its exhaustiveness and any possible missed data is important to know. The ethical committee

Line 86 : are there any other reasons for confusion factors? Do we have data on coding accuracy?

Line 88: Did the authors check for a more general coding Like R99?

Lines 111-119: definition of cumulative and peak use of care would fit better in the outcomes section. I would like the authors to clarify the discrepancy between primary care and specialist care representation where they included the first visit only for primary care while they calculated the hospital bed days.

Lines 126-135: I suggest the overall results being presented first before going into the different subgroups analysis to make the reading easier and to be in accordance with the primary objective of the study and the analysis plan described in the statistical analysis section

Lines 157-159: I couldn’t see on this figure that the share in need of specialist care prior to primary care was larger than its inverse. It may need review and clarification.

Death data was not presented in the results section

the term "Share" and "Fraction" are used mutually in different locations creating confusion. i suggest to authors using one term throughout the article.

The discussion is very well structured. It states the main results with a global view on the objectives of the study and discuss the possible reasons of the observed trends while suggesting hypotheses to test in future research. The limitations are well discussed making the conclusions drawn from this exploratory analysis reasonable and sound.

Figures are difficult to read because of the low resolution. Y axis unit is not shown for fig 1,2 and 4

The design of figure 1 makes the comparison of the different sex and age groups not easy.

The S1 Fig shows the absolute numbers on the y axis which makes any visual comparison erroneous.

6. PLOS authors have the option to publish the peer review history of their article (what does this mean?). If published, this will include your full peer review and any attached files.

Reviewer #1: No

Reviewer #2: **Yes: **Benjamin Tolchin

Reviewer #3: **Yes: **Marouan Zoghbi

---

## [Author Response · Author response to Decision Letter 0]

2 Mar 2022

(For more intuitive formatting see the uploaded file with response to reviewers.)

We would like to thank the expert reviewers for valuable input, which has helped to improve the quality of our manuscript. Please find below a point-to-point response to your comments and a list of the changes we made in the revised manuscript. 

Comments of Academic Editor

1-Describe the study design, make sure your paper follows the STROBE checklist for observational studies

We agree. We incorporate a sentence where we describe the study design as a national cohort study using registry data, p. 3, lines 50-52: “Thus, in this paper we aim to explore the age- and sex-specific acute burden of COVID-19 on the health care services in four waves of the pandemic in Norway through a national cohort study using registry data.”We have also attached a STROBE checklist.

2-Please address the comments of reviewer #2 regarding the BEREDT-C19 registry and its validation data. What are the flaws of the registry, missing data? 

We agree that issues such as flaws of the registry including missing data should have been better described. We have now described the flaws of the registry (such as non-quality checked data) in the methods section, p. 4-5 and lines 77-80: “Overall, data from Norwegian health registers have been demonstrated to be of high quality with high validity and reliability, and together they can provide a complete picture of patterns of healthcare use [11-13]. Medical recording to the National registries is mandated by law in Norway, ensuring no missing data in our study.”

3-Clearly define the criteria for primary care and specialist care, as highlighted by reviewers #2 and #3. 

Yes, the definitions of primary care and specialist visits could have been clearer. To make this clearer to the reader we have changed from “primary care” to “outpatient care”, and from “specialist care” to “inpatient care”, as suggested by the reviewers.

4-Please explain the rationale behind using age- and sex-specific share of patients to assess healthcare system use and the rationale behind the 30-day acute phase as highlighted by reviewer #3 

We agree this could have been better described in the background section. Age and sex differences were already well-known, e.g. that children are less severely affected than elderly in terms of hospital admissions. Here, we aimed to shed light on admissions in combination with primary care visits for the different age and sex groups. We have added the following to the introduction, p. 3-4, lines 52-56): “Because of previously reported differences in vaccination status throughout the pandemic [7], differences in disease severity by SARS-CoV-2 variant [8, 9] and strata of age and sex [10], we hypothesized that we would also see differences in the 30-day pattern of healthcare use for men and women, girls and boys, the working age population and the elderly in the different waves of the pandemic.” And the following sentence to the methods section, p. 5, lines 87-90: “We divided our population into mutually exclusive age and sex groups, i.e. girls and boys, men and women by the following age categories: 1-19 (children and adolescents), 20-67 (working age population) and 68 years or older (elderly), as COVID-19 has hit differently among different age groups and sexes [10].” And the following to the methods section, p. 6, lines: 101—104: “We chose a 30-days-timeframe because a death after COVID-19 was classified as covid-related if it occurred within 30 days after testing positive in official statistics [15], and because it coincides with what is commonly referred to as the acute phase of SARS-CoV-2 [16].”

5-Clarify figure S1 

We agree. We have now removed the numbers on the Y-axis in Figure S-1 to remove any confusion, and we have added more information in the figure notation.

6-Please use one term either share or fraction 

We agree. We have revised our manuscript throughout to use the term “share” instead of “fraction”.

Comments of reviewer 1

Thanks to the authors. I reviewed the article. It is valuable research in the pandemic era. The title, introduction, methodology, and conclusion are appropriate. There is no plagiarism in it. The writing is appropriate and completely understandable. I have minor recommendation as followings: 

Thank you for your encouraging comments. 

1. How did the authors delete the effect of confounding factors from the study outcome?

In our study, we aimed to simply describe the health services use in primary and specialist care (now called outpatient and inpatient care) following a positive test for SARS-CoV-2. We did not aim to test a causal hypothesis, and thus, there was no need to control for any potentially confounding factors. However, we agree that factors impacting on health care use, such as SARS-CoV-2 variant and vaccination could have been described in our paper, although there was no need to adjust for them in the statistical analyses. Besides more clearly describing that we had a descriptive aim and providing a clearer background for a broad hypothesis of what we expected to find in the descriptive data, we now include more data on SARS-CoV-2 variants and vaccination grade, please see Figure 1. We also include more of this information in our discussion section. 

2- Discussion should be improved and revised based on the study results and comparison with previous ones i.e. Bagi HM, Soleimanpour M, Abdollahi F, Soleimanpour H (2021) Evaluation of clinical outcomes of patients with mild symptoms of coronavirus disease 2019 (COVID-19) discharged from the emergency department. PLoS ONE 16(10): e0258697. https://doi.org/10.1371/journal.pone.0258697

Thank you for this valuable input. We have included a more thorough discussion and included the suggested references. We have also improved the structure of our entire discussion section, with headings “Comparison to previous studies”, “Interpretation and relevance” and “Strengths and limitations” (edits too long to be pasted here). 

3- What is the new finding of this study compared to previous ones this field? 

We agree that we could have better described the independent contribution relative to previous studies. Amongst others, we have made the following revisions to our discussion section, p. 13, lines 252-264: 

 “To our knowledge, this study is the first to describe COVID-19 related patterns in both inpatient and outpatient care use following positive test for SARS-CoV-2. The massive use of outpatient care services (including visits to the general practitioner and emergency wards) following COVID-19 has not previously been reported, i.e. no knowledge foundation exists for an effective comparison of findings across studies. Far more is known for inpatient care use. Compared to previous studies we find, on average, lower hospitalization rates in Norway than what has been reported in other countries [20-22]. As an example, Denmark and Sweden observed hospitalization rates of 20% and 16%, respectively, in the first wave of the pandemic [21, 22], which contrasts with Norwegian hospitalization rates of only 5% (Table 1). The differences may be explained by differences in test criteria, but also differences in criteria for admission to the hospital. Further, the Danish and Swedish rates are based on a pandemic period that was shorter than studied in the current study [21, 22]. There were also differences in mortality between the Scandinavian countries.”

4- Please mention the weak and strong points of your study 

We agree that the weak and strong points of our study could have been better structured. Please see our section of the discussion section entitled “Strengths and limitation”, p. 15-17, lines 308-342. 

5- The list of abbreviations at the end of the manuscript is absence; please make sure all your abbreviations are listed. 

We see that this could be useful, however we could not find that this is standard practice in this journal. We will add a list of abbreviations if required by the journal. 

6- I recommend providing a table on the characteristics of your sample, including all variables you use in your analysis later on. 

We agree. We have enlarged Table 1 with more information, and we now also include a new table (S1-Table) in the appendix showing the numbers and percentages of all health care use for each sub-groups and waves.

7- Please describe the study design in more detail. 

We agree. We have added the following to the introduction section, p. 3, lines 50-52: “Thus, in this paper we aimed to explore the age- and sex-specific acute burden of COVID-19 on the health care services in four waves of the pandemic in Norway through a national descriptive cohort study design using registry data.”

8- provide more information on how you analyzed your data and possibly provide some references for e.g. level of statistical significance. 

We agree. We have added the following to the statistical analyses section, p. 8, lines 138-140: “For each age-/sex-group and for each of their pathways in each of their pandemic wave, we estimated the proportions having the different pathways and calculated 95% confidence intervals.” Accordingly, we have updated our Figure 6 with the confidence intervals. 

Comments of reviewer 2

Introduction (pages 3-4): Please state study design (national cohort study?) and any prespecified hypotheses. More generally, please review and consider using STROBE checklist for observational studies (e.g. https://www.strobe-statement.org/checklists/). 

Thank you. We have added the following to the introduction section, p. 3, lines 50-52: “Thus, in this paper we aimed to explore the age- and sex-specific acute burden of COVID-19 on the health care services in four waves of the pandemic in Norway through a national descriptive cohort study design using registry data.” Our study design was of a descriptive and explorative nature, and we had no prespecified hypothesis in terms of direction of results. However, as rationale for our choice of groupings and categorizations, we have added the following hypotheses, p. 3-4, lines 52-56: “Because of previously reported differences in vaccination status throughout the pandemic [6], differences in disease severity by SARS-CoV-2 variant [7, 8] and strata of age and sex [9], we hypothesized that we would also see differences in the 30-day pattern of healthcare use for men and women, girls and boys, the working age population and the elderly in the different waves of the pandemic.” We have also attached a STROBE checklist.

Methods (page 4): An important feature of this study is its use of Norwegian national registries. The authors give one reference to a webpage that describes the BEREDT-C19 registry. I’m unable to find any validation data on that website. If such validation data exists, the authors should cite it directly. If validation data of the BEREDT-C19 registry does not exist but validation of the underlying registries from which BEREDT-C19 draws does exist, I would cite that data instead. The validity of these registries may be well known to Norwegian epidemiologists, but should be established for international readers. 

Thanks for pointing out. We agree that this would greatly benefit the paper. We now include a sentence on the quality of the underlying data of the register, together with a reference. See methods section, p. 4-5 and lines 77-80: “Overall, data from Norwegian health registers have been demonstrated to be of high quality with high validity and reliability, and together they can provide a complete picture of patterns of healthcare use [11-13]. Medical recording to the National registries is mandated by law in Norway, ensuring no missing data in our study.”

Methods (page 5-6): It becomes apparent here that the authors are including treatment received in emergency wards as “primary care,” as contrasted with “specialist care” meaning hospital admission. I personally wouldn’t usually think of emergency department visits as part of primary care, and I don’t think most US physicians would either. Conversely, specialist care doesn’t clearly connote hospitalization to me (I would think of e.g. an outpatient cardiology clinic visit as “specialist care”). If I’m understanding the authors’ intentions correctly, I wonder if the manuscript might be clearer for international readers if it consistently referred to “outpatient care” (including clinic and emergency department visits) vs. “inpatient care” (hospital admission) rather than to “primary care” vs. “specialist care?” 

Thank you. We agree. We now use the term “outpatient care” instead of “primary care”, and “inpatient care” instead of “specialist care”.

Methods (page 6): the authors describe the three waves they are examining and mention the start of mass vaccination during the third wave. It would be helpful to have numbers and citations for the rates of vaccination at the beginning and end of this wave, since this would be expected to significantly impact rates of hospitalizations and I would think this data would be relatively easy to obtain. 

We agree. We now incorporate a graph showing the share of persons vaccinated with at least one dose (Fig 1). Moreover, we include this in the Discussion, see p. 15, lines 300-306: “The milder SARS-CoV-2 omicron variant and a higher vaccination coverage might also explain the observed shifts in health care use (Fig 1). However, we cannot exclude that more severe mutations in the future again place an increasing demand on the healthcare services. Also, vaccines might have lower effects against new variants. Thus, if we again see SARS-CoV-2 variants that cause the same disease severity as the initial variants, our findings of an only halved outpatient care use but ten times lower inpatient care use from March 2020 to February 2022 suggest that an upscaling of the outpatient care services might be particularly important in the future.” 

Results (pages 7-11): The authors don’t have a comparison group but make many comparisons between waves, e.g. “we observed a significant shift in the total use of specialist care throughout the pandemic, from 14% being hospitalized at least once during the 1st wave, compared to 4% during the 2nd and 3rd waves (Fig 4).” (page 10) Consider measures of statistical significance or confidence intervals for differences

We agree. We now include a table (S-Table 1) showing all rates with corresponding 95% confidence interval, to show whether the differences in health care use across the various wave are significant or not. We have also updated Fig 6 showing 95% confidence intervals.

Discussion (page 12): The authors state that “we were unable to find a similar study for an effective comparison of our findings.” I agree that I also am not able to find a similar study of the proportion of COVID patients receiving outpatient care. However, as the authors note, there are many studies of hospitalization rates among patients with COVID (e.g. Menachemi N, Dixon BE, Wools-Kaloustian KK, Yiannoutsos CT, Halverson PK. How Many SARS-CoV-2-Infected People Require Hospitalization? Using Random Sample Testing to Better Inform Preparedness Efforts. J Public Health Manag Pract. 2021 May-Jun 01;27(3):246-250. doi: 10.1097/PHH.0000000000001331. PMID: 33729203.). The authors should discuss how their findings regarding hospitalization rates compare to those found in other studies (the hospitalization rates the authors found appear to me higher than those found in some other studies).

Thank you for suggesting relevant literature. We now include a more thorough discussion on how our findings relate to previous literature on COVID-19 patients seeking inpatient health care. We include the provided references. Please see our response and action to the comments above and our new section included in the manuscript, entitled “Comparison with previous studies”, p. 13-14 (too lengthy to be pasted here). 

Discussion (page 13): The authors note that the proportion of patients hospitalized and the duration of hospitalization increased from wave 2 to wave 3. Given that the authors state that vaccines became available during wave 3, these findings are surprising. Again, it would be helpful to know what proportion of the population was vaccinated at the beginning and end of the 3rd wave. Some comment might be offered in the discussion. 

We agree that showing the proportion of vaccinated would improve the paper. We now include a graph (Fig 1) showing the share of the population vaccinated with at least one dose throughout the various waves. We also include the following section in the discussion section, p. 14, lines 271-278: "Another important observation to our inpatient care results, was the tendencies that the mean number of bed-days in hospital increased for some age- and sex-groups from the 1st to the 2nd wave (Fig 5). Although we did not aim to explore whether the severity of COVID-19 has changed, an important characteristic of the 2nd, 3rd, and 4th waves of transmission has been the rise of mutant viruses. Reports have been inconclusive as to whether mutant viruses (Alpha/Beta/Gamma vs. Wuhan strain) result in more severe disease requiring more hospital care [19, 24, 25]. Whilst the risk of hospitalization between the Alpha and Delta variant were found similar, the Omicron variant has had a reduced risk of hospitalization compared to the Delta variant [8, 9].”

Finally, the authors may consider including data from the current omicron wave. I don’t think it’s absolutely necessary, and the publication need not and should not be delayed until the pandemic fully passes, but the study would be even more informative with data from this fourth wave included. 

We agree that this would greatly improve the paper. We have now extended the study period until 14th of February 2022 (last positive test on 14th of January 2022), to capture data on the latest wave. 

Comments of reviewer 3

This study highlights a major topic of interest during the Covid 19 pandemic. The healthcare burden imposed by the pandemic is one of the main factors to be studied in order to improve preparedness plans for the coming waves and to enhance knowledge about dealing with future pandemics. This study deals with exhaustive data from a national registry and is theoretically well positioned to give valuable information about the research question risen by the authors.

Understanding patients flow during the pandemic is another important facet of managing healthcare resources knowing that the study reports data from Norway, a country known for a very distinguished and organized health system.

The authors decided to study sex and age specific use of healthcare system, I would appreciate adding to the introduction the rationale behind this choice as they were only mentioned in the research question without any evidence to support this choice.

Thank you for your encouraging feedback. We decided to study sex and age specific use as healthcare use have differed between the two sexes and for various age groups. A 30-day frame have often been considered as the acute phase. Anything beyond that is often considered as “long covid”, which is not the topic of this article. We agree that the rationales for our choices could have been better described. We have added the following to the introduction, p. 34, lines 52-56): “Because of previously reported differences in vaccination status throughout the pandemic [6], differences in disease severity by SARS-CoV-2 variant [7, 8] and strata of age and sex [9], we hypothesized that we would also see differences in the 30-day pattern of healthcare use for men and women, girls and boys, the working age population and the elderly in the different waves of the pandemic.” And the following sentence to the methods section, p. 5, lines 87-90: “We divided our population into mutually exclusive age and sex groups, i.e. girls and boys, men and women by the following age categories: 1-19 (children and adolescents), 20-67 (working age population) and 68 years or older (elderly), as COVID-19 has hit differently among different age groups and sexes [9].”

The 30 days’ time frame is also a choice made by the authors and showing evidence supporting this time limit -mean time to recovery, incidence of late onset complications…- would also be of value. 

We agree. We have added the following to the introduction, p. 3-4, lines 52-56): “Because of previously reported differences in vaccination status throughout the pandemic [6], differences in disease severity by SARS-CoV-2 variant [7, 8] and strata of age and sex [9], we hypothesized that we would also see differences in the 30-day pattern of healthcare use for men and women, girls and boys, the working age population and the elderly in the different waves of the pandemic.” And the following to the methods section, p. 6, lines: 101—104:“We chose a 30-days-timeframe because a death after COVID-19 was classified as covid-related if it occurred within 30 days after testing positive in official statistics [15], and because it coincides with what is commonly referred to as the acute phase of SARS-CoV-2 [16]. Thus, we regarded people who were still alive after 30 days as recovered.”

The authors developed in a descent way the importance of their research question. Developing the potential applications of such data would add value to the manuscript 

We agree. We now incorporate a sentence on what the data may be used to in other studies, on p. 3, lines 43-48: “A timely and correct up- and downscaling of health services (and lockdown measures) depend on our understanding of the pathways patients take through the health system, including the peaks and total demand of health care services following an individual’s positive test. Studying the impact on both inpatient and outpatient care in its early waves, can provide valuable insight into health service needs in later stages, and contribute to the knowledge base that can increase our resilience against future pandemics.” These and other revisions contributed to a significantly increased word count of our introduction section, i.e. we have also deleted the two first sentences describing lockdown (line 36-40) as we think it was less relevant for our research question. 

The outcomes measures are not well described in the methods section. Primary care use for example is a broad topic and the indicators used for measurement developed later may figure in a paragraph with a detailed description of the calculation methods used for every variable used to assess this outcome.

We agree. We now use the term ‘outpatient’ and ‘inpatient’ instead of ‘primary’ and ‘specialist’ care, respectively. We have also restructured our outcomes section, placing the specific outcomes first, and how they were handled in the analyses thereafter. 

A review of the editing and English proofing would also be helpful to make the reading even more enjoyable. 

We agree. We have edited the English language. 

Lines 59-72: the description of the registry is fair enough and gives a good insight of its contents and objectives. A clarification about its exhaustiveness and any possible missed data is important to know. The ethical committee 

We agree. We have added the following to p. 4-5, lines 77-80: “Overall, data from Norwegian health registers have been demonstrated to be of high quality with high validity and reliability, and together they can provide a complete picture of patterns of healthcare use [11-13]. Medical recording to the National registries is mandated by law in Norway, ensuring no missing data in our study.” 

Line 86 : are there any other reasons for confusion factors? Do we have data on coding accuracy?

In our study, we aimed to simply describe the health services use in primary and specialist care (now called outpatient and inpatient care) following a positive test for SARS-CoV-2. We did not aim to test a causal hypothesis, and thus, there was no need to control for any potentially confounding/confusion factors. However, we agree that factors impacting on health care use, such as SARS-CoV-2 variant and vaccination could have been described in our paper, although there was no need to adjust for them in the statistical analyses. We agree that we could have included more information regarding coding accuracy. With regards to confusion factors, we have provided a better description of our study design as well as a rationale for our research aim and hypothesis, please see the introduction section, p. 3-4, lines 52-56. Further, we provide data on factors impacting on health care use, such as vaccination and SARS-CoV-2 variant in Fig 1. Regarding coding accuracy, please see response above.

Line 88: Did the authors check for a more general coding Like R99?

The more general code R99 was considered, however as it represents any “respiratory tract infection not classified elsewhere”, i.e. not restricted to covid-related healthcare use, we decided to avoid it in our study. 

Lines 111-119: definition of cumulative and peak use of care would fit better in the outcomes section. I would like the authors to clarify the discrepancy between primary care and specialist care representation where they included the first visit only for primary care while they calculated the hospital bed days.

We agree that peak and total use of healthcare could have been described in the Outcomes section. However, we would like to keep the details of our estimation strategies for the peak and total healthcare use in the statistical analyses section. We have added the following to the section on outcomes, page 6, lines 105-107: “When combined, and sorted chronologically on dates of occurrence, these data provided a comprehensive picture of the peak and total use of outpatient and inpatient care, as well as COVID-19-related health care pathways in the acute phase.” Moreover, we now include all outpatient care visits when estimating peak and total use, not only the first visit. This also makes it more comparable to the way we measure inpatient care, as we include all bed-days and not just the admission date.

Lines 126-135: I suggest the overall results being presented first before going into the different subgroups analysis to make the reading easier and to be in accordance with the primary objective of the study and the analysis plan described in the statistical analysis section 

Thank you for the suggestion. We agree. We now present the overall results before presenting the different subgroup analyses (patient pathways). We have revised the chronology of the statistical analyses section accordingly. 

Lines 157-159: I couldn’t see on this figure that the share in need of specialist care prior to primary care was larger than its inverse. It may need review and clarification. We believe that it is even clearer now. In S1 Fig, when looking at the top right figure, the share going from inpatient care (IC) to outpatient care (OC) is larger than the share going from outpatient care (OC) to inpatient care (IC).

Death data was not presented in the results section.

Death data were presented in the first paragraph in the Results section. However, we agree that these data should be described more thoroughly (lines 149-152). We now present even more death data by the inclusion of all-cause mortality in Table 1 and more in-depth in S1 Table. We also describe mortality more thoroughly in the Discussion, p. 3, lines 264-270: “Whereas all-cause mortality among COVID-19 patients in Sweden was observed to be around 4-5% [22], we find a mortality rate of only 0.9% in Norway between February 2020 and February 2021 (Table 1). Again, the discrepancies might be explained by the difference in length of the study periods as well as differences in registration practices of deaths. However, in line with previous studies, we see higher mortality rates among men compared to women, particularly for those younger than 68 years [10]. Also, in line with previous reports, we find that a significant share of those who died, died within the first 10 days (S1 Fig) [23].”

the term "Share" and "Fraction" are used mutually in different locations creating confusion. i suggest to authors using one term throughout the article.

We agree. We now use the term “share” throughout the paper.

The discussion is very well structured. It states the main results with a global view on the objectives of the study and discuss the possible reasons of the observed trends while suggesting hypotheses to test in future research.

Thank you. In accordance with comments from Reviewer 1 and 2, we have improved our discussion section even further, by the inclusion of subheadings, such as “Comparison to previous studies”, “Interpretation and relevance”, and “Strengths and limitations”.

The limitations are well discussed making the conclusions drawn from this exploratory analysis reasonable and sound.

Thank you.

Figures are difficult to read because of the low resolution. Y axis unit is not shown for fig 1,2 and 4 

Thanks for letting us now. Figures are now in higher quality. 

The design of figure 1 makes the comparison of the different sex and age groups not easy

We agree that the figure layout could have been better. We have restructured the figure (now renamed Fig 6) in such a way that the periodical shifts in healthcare use are now visually evident. We have also attached the 95% confidence intervals. 

The S1 Fig shows the absolute numbers on the y axis which makes any visual comparison erroneous.

We agree. We have now removed the numbers from the y-axis, so that it is easier to understand that comparison is of percentages and not absolute numbers, and we have also added more information to the figure notation.

---

## [Editor Report · Decision Letter 1]

4 Mar 2022

PONE-D-21-37160R1Pandemic trends in health care use: From the hospital bed to self-care with COVID-19PLOS ONE

Dear Dr. Methi,

Thank you for submitting your revised manuscript to PLOS ONE. It still needs a minor revision.Please remove the figure 1 from the introduction and integrate it within a paragraph entitled "study setting" in the methods, where you can describe in a sentence or two the SARS-CoV-2 context in Norway.

We look forward to receiving your revised manuscript.

Kind regards,

Mabel Aoun, MD, MPH

Academic Editor

PLOS ONE
---

## [Author Response · Author response to Decision Letter 1]

7 Mar 2022

Please remove the figure 1 from the introduction and integrate it within a paragraph entitled "study setting" in the methods, where you can describe in a sentence or two the SARS-CoV-2 context in Norway. 

We agree. We removed Fig 1 from the introduction and placed it within a subsection entitled “Study setting” within the Methods section. Instead of adding one to two sentences on the SARS-CoV-2 context in Norway, we moved the paragraph describing the different waves from the subsection “Statistical analyses” to this new section, as we felt it was more logical to place the paragraph in the new subsection.

We also noticed that we had made an error on line 92. We do not only include hospitalizations with ICD-10 code U071 (confirmed COVID-19), but we also include U072 (suspected COVID-19). This is because these patients have already tested positive for SARS-CoV-2, and in the beginning of the pandemic, hospitals were not coherent in in the use of U071 or U072. This is now corrected in the uploaded manuscript.

---

## [Editor Report · Decision Letter 2]

9 Mar 2022

Pandemic trends in health care use: From the hospital bed to self-care with COVID-19

PONE-D-21-37160R2

Dear Dr. Methi,

We’re pleased to inform you that your manuscript has been judged scientifically suitable for publication and will be formally accepted for publication once it meets all outstanding technical requirements.

Kind regards,

Mabel Aoun, MD, MPH

Academic Editor

PLOS ONE
---

## [Editor Report · Acceptance letter]

14 Mar 2022

PONE-D-21-37160R2 

Pandemic trends in health care use: From the hospital bed to self-care with COVID-19 

Dear Dr. Methi:

I'm pleased to inform you that your manuscript has been deemed suitable for publication in PLOS ONE. Congratulations! Your manuscript is now with our production department. 

Kind regards, 

on behalf of

Dr. Mabel Aoun 

Academic Editor

PLOS ONE